



# Tropospheric NO₂, SO₂, and HCHO over the East China Sea, using ship-based MAX-DOAS observations and comparison with OMI and OMPS satellites data

Wei Tan[1], Cheng Liu[1,2,3,6,*], Shanshan Wang[4,5,*], Chengzhi Xing[2], Wenjing Su[2], Chengxin Zhang[2], Congzi Xia[2], Haoran Liu[2], Zhaonan Cai[7], Jianguo Liu[1]

[1]Key Lab of Environmental Optics and Technology, Anhui Institute of Optics and Fine Mechanics, Hefei Institutes of Physical Science, Chinese Academy of Sciences, Hefei, 230031, China

[2]School of Earth and Space Sciences, University of Science and Technology of China, Hefei, 230026, China

[3]Center for Excellence in Regional Atmospheric Environment, Institute of Urban Environment, Chinese Academy of Sciences, Xiamen, 361021, China

[4]Shanghai Key Laboratory of Atmospheric Particle Pollution and Prevention (LAP³), Department of Environmental Science and Engineering, Fudan University, Shanghai, 200433, China

[5]Shanghai Institute of Eco-Chongming (SIEC), No.3663 Northern Zhongshan Road, Shanghai, 200062, China

[6]Anhui Province Key Laboratory of Polar Environment and Global Change, USTC, Hefei, 230026, China

[7]Key Laboratory of Middle Atmosphere and Global Environment Observation, Institute of Atmospheric Physics (IAP), Chinese Academy of Sciences, Beijing 100029, China

*Correspondence to*: Shanshan Wang (shanshanwang@fudan.edu.cn), Cheng Liu (chliu81@ustc.edu.cn)

**Abstract.** In this study, ship-based Multi-Axis Differential Optical Absorption Spectroscopy (MAX-DOAS) measurements were performed in the Eastern China Sea (ECS) area in June 2017. The tropospheric Slant Column Densities (SCDs) of nitrogen dioxide (NO₂), sulfur dioxide (SO₂), and formaldehyde (HCHO) were retrieved from the measured spectra by the Differential Optical Absorption Spectroscopy (DOAS) technique. Using the simple geometric approach, the SCDs of different trace gases observed at 15° elevation angle were adopted to convert into tropospheric Vertical Columns Densities (VCDs). During this campaign, the averaged VCDs of NO₂, SO₂, and HCHO in the marine environment over ECS area are $6.50 \times 10^{15}$ molec cm⁻², $4.28 \times 10^{15}$ molec cm⁻² and $7.39 \times 10^{15}$ molec cm⁻², respectively. In addition, the ship-based MAX-DOAS trace gases VCDs were compared with satellite observations of Ozone Monitoring Instrument (OMI) and Ozone Mapping and Profiler Suite (OMPS). The daily OMI NO₂ VCDs agree well with ship-based MAX-DOAS measurements showing the correlation coefficient R of 0.83. Besides, the good agreements of SO₂ and HCHO VCDs between the OMPS satellite and ship-based MAX-DOAS observations were also found with correlation coefficient R of 0.76 and 0.69. The vertical profiles of these trace gases are achieved from the measured Differential Slant Column Densities (DSCDs) at different elevation angles using optimal estimation method. The retrieved profiles displayed the typical vertical distribution characteristics, which exhibits the low concentrations of < 3, < 3, and < 2 ppbv for NO₂, SO₂, and HCHO in clean area of the marine boundary layer





far from coast of the Yangtze River Delta (YRD) continental region. Interestingly, elevated $SO_2$ concentrations can be observed intermittently along the ship routes, which is mainly attributed to the vicinal ship emissions in the view of the MAX-DOAS measurements. Combined with the on-board ozone lidar measurements, the ozone ($O_3$) formation was discussed with the vertical profile of HCHO/$NO_2$ ratio, which is sensitive to the increases of $NO_2$ concentration. This study provided further understanding of the main air pollutants in the marine boundary layer of the ECS area and also benefited to formulate the

policies regulating the shipping emissions in such costal area like YRD region.

## 1    Introduction

Nitrogen dioxide ($NO_2$), sulfur dioxide ($SO_2$) and formaldehyde (HCHO) are the important atmospheric trace gases which play a major role in the atmospheric chemical processes. $NO_2$ participates in the formation of ozone ($O_3$) and reacts with hydroxyl radicals (OH), the strongest oxidizing agent in the atmosphere, to produce aerosols and acid rain, which are harmful to both

buildings and human health (Seinfeld and Pandis, 2006; Lelieveld and Dentener, 2000; Lelieveld et al., 2002). $NO_2$ may also has important impacts on the greenhouse effect (Solomon et al., 1999). Besides natural sources, high-temperature combustion processes, e.g. fossil fuel burning, accidental and intentional biomass burning, are estimated to contribute the major emissions of nitrogen oxides ($NO_x=NO_2+NO$) (Lee et al., 1997). $SO_2$ contributes to the formation of sulfate aerosols and acid rain, both of which have negative effects on the climate and human health, as well as lead to building acid corrosion (Hutchinson and

Whitby, 1977; Pope and Dockery, 2006; Longo et al., 2010). The dominant anthropogenic emissions of $SO_2$ are the burning of fossil fuels, smelters, and oil refineries, whereas the discharge of active volcanoes is the major natural source. HCHO is the predominant product of the oxidation of many Volatile Organic Compounds (VOCs) by OH radical and is abundant throughout the atmosphere. Therefore, elevated HCHO levels can be related to the emission of reactive Non-Methane Volatile Organic Compounds (NMVOCs) originating from biogenic, pyrogenic or anthropogenic sources (Fu et al., 2007; Millet et al., 2008;

Stavrakou et al., 2009a; 2009b).

The Differential Optical Absorption Spectroscopy (DOAS) technique is widely used to identify and quantify kinds of the atmospheric trace gases. Based on the DOAS principle, the quantitative of the trace gases was acquired from the narrow band absorption structures of the different trace gases, which were separated from the broad band parts caused primarily by the atmospheric scattering and their broad band absorption (Platt and Stutz, 2008). The named Multi-AXis-Differential Optical

Absorption Spectroscopy (MAX-DOAS) instrument is designed to observe scattered sunlight under different viewing angles closed to the horizontal and the zenith direction, which can provide high sensitivity to tropospheric aerosols and trace gases (Hönninger et al., 2004). In the past decades, the MAX-DOAS method has been successfully used for many atmospheric trace gases observation on different platforms, such as $NO_2$, $SO_2$, HCHO, HONO and others. The most common application is ground based measurement (e.g. Irie et al., 2011; Pinardi et al., 2013; Wang et al., 2014; Chan et al., 2015; Xing et al., 2017).



Meanwhile, the mobile platform observations has been developed rapidly, e.g. car-based observations (Johansson et al., 2008;

Shaiganfar et al., 2011; Wang et al., 2012; Shaiganfar et al., 2017), aircraft-borne observations (Baidar et al., 2013; Dix et al.,

2016), and ship-based observations (Sinreich et al., 2010; Takashima et al., 2012; Peters et al., 2012; Schreier et al., 2015;

Hong et al., 2018).

Usually, the trace gases concentrations are very low in remote marine environments considering there are no emission sources

except the ship traffic and some other natural sources. Previous ship-based MAX-DOAS studies reported that $NO_2$ Vertical

Column Densities (VCDs) were basically low ($< 0.50 \times 10^{15}$ molec cm$^{-2}$) due to the absence of obvious $NO_x$ emission sources

nearby, and the $NO_2$ concentration in marine boundary layer extracted from profile retrieval are $< 30$ pptv in the open and

clean tropical sea area of the South China and Sulu Sea (Schreier et al., 2015). Times series of $SO_2$ magnitudes were found to

be consistent with tropospheric $NO_2$ over there and occasionally increased if the measurements taken in a busy shipping lane.

Over western Pacific and Indian Ocean area, the background value of $NO_2$ concentration was less than $\sim 0.2$ ppbv over the

remote ocean (Takashima et al., 2012). Peters et al. (2012) found that HCHO VCDs over the remote ocean exhibit a diurnal

pattern with maximum values of $4 \times 10^{15}$ molec cm$^{-2}$ at noontime over the western Pacific Ocean, and corresponding retrieved

peak concentrations were up to 1.1 ppbv at higher altitudes around 400 m. However, these pollutants concentrations increased

to a high value when the measurements were taken close to the shore, busy ports, or vessels (Takashima et al., 2011; Peters et

al., 2012; Schreier et al., 2015). So far, the air quality of marine boundary layer in the China coastal area are rarely reported.

In this study, we used the ship-based MAX-DOAS measurements to report the column densities and temporal-spatial

distributions of $NO_2$, $SO_2$, and HCHO in the marine environments over the East China Sea (ECS) area in June 2017. During

this campaign, the cruise ship mainly navigated at the sea area surround Yangtze River Delta (YRD) region, which is the

confluence of the coastal shipping routes and inland water transportation on the Yangtze River. It is the busiest waterways of

the ECS area and also the one of the three key Ship Emission Control Zones (ECZs) of China. The YRD coastal port cluster

is composed of >15 ports, of which Shanghai and Ningbo-Zhoushan have served as the largest two container ports in the world

since 2013. With the flourish shipping industry, the throughputs of YRD ports strikes the new highs continuously and caused

the considerable ship emissions of $SO_2$, $NO_x$ and $PM_{2.5}$, which has significant impacts to the local and regional air pollution in

both offshore and inland area of YRD region (Fan et al., 2016; Zhang et al., 2017). Due to the rapid development of urbanization

and industrialization, as well as expanding population, the continental YRD region also suffered from the ecological

degradation and environmental problems at the same time, e.g. atmospheric fine particle and $O_3$ pollution (Chen et al., 2017;

Song et al., 2017).

In this paper, both the VCDs and vertical distributions of $NO_2$, $SO_2$ and HCHO from ship-based MAX-DOAS measurements,

as well as the ozone profiles from on-board lidar, have been firstly reported for the ECS area covering the Yangtze River

Estuary and surrounding YRD region waters. These observed data sets are vital for the better understanding of the air quality





in marine boundary layer along the coastline of China and helpful for regulating the air pollution in coastal area.

## 2    Methodology

### 2.1    The cruise of ship-based observation

The ship-based measurements campaign was implemented over the offshore marine area of the ECS covering the Yangtze

River Estuary and YRD region coast in summer from 2 to 29 June, 2017 (Fig. 1(a)). Before the departure, measurement

instruments were installed and debugged at Gongqing port (point A in Fig. 1(b), 31.33° N, 121.55° E) of Shanghai on 1 June

2017. As indicated in Fig. 1(b), the ship set sail on 2 June 2017 from Gongqing port by way of Yangshan port (B, 30.62° N,

122.09° E), and Daishan islands (C, 30.25° N, 122.16° E) and Hagnzhou Bay area. After encircled Zhoushan islands (29.99°

N, 122.20°), the ship moved forward to Shengsi islands (D, 30.71° N, 122.45° E) and Huaniao islands (E, 30.85° N, 122.68°

E). Then, the ship was heading to Lianxing port in Jiangsu Province (F, 31.72° N, 121.87° E) and passing through Huaniao

islands finally back to Gongqing port on 29 June, 2017. As shown in Fig. 1 the ship cruise routes not only covered the busy

waters in YRD region but also pass though some clean marine area 100 km away from the continental coast.

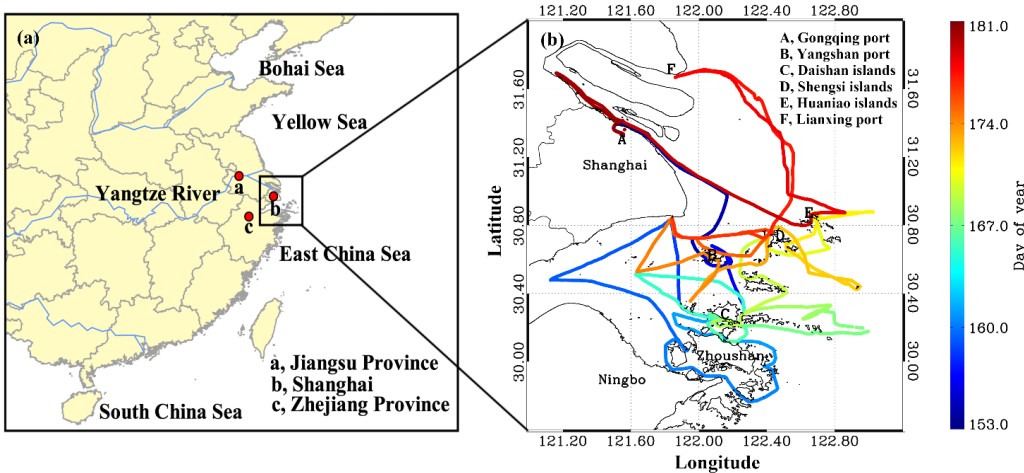

**Figure 1.** Location (a) and cruise routes (b) of the ship-base campaign from 2 to 29 June, 2017 (DOY 153 to 181).

**2.2    Ship-based MAX-DOAS measurements**

### 2.2.1 Instrument setup

An integrated and fully automated MAX-DOAS instrument was fixed on a 1.5 m height tripod top at the stern deck of the ship.

This compact instrument consists of an ultraviolet spectrometer (AvaSpec-ULS2048L-USB2) covering the spectral range of

300-460 nm with a spectral resolution of 0.6 nm, a one-dimensional CCD detector (Sony ILX511, 2048 individual pixels) and

a stepper motor driven the telescope to collect scattered sunlight from different elevation angles (angle between the horizontal



and the viewing direction, $\alpha$). Besides, the controlling electronic devices, connecting fiber, and other necessary devices are mounted inside too. To avoid the impact of emission plums from the ship itself, the azimuthal angle of the telescope unit kept at 130º relative to the heading direction of the observation ship. The telescope was scanning in the sequence of elevation angles of 3º, 5º, 7º, 10º, 15º, 30º, and 90º. The duration of an individual spectrum measurement was about 30 s and the each scanning

sequence took about 4 min. The daily measurements were automatically controlled by a built-in computer combine with a spectral collection software when the Solar Zenith Angle (SZA) less than 75°. Moreover, a high-precision Global Position System (GPS) data receiver was configured to record the real-time coordinate positions and the track the ship cruise.

**2.2.2 Spectral analysis**

Based on the DOAS principle, the measured scattered sun-light spectra are analyzed using the QDOAS spectral fitting software

suite developed by BIRA-IASB (http://uv-vis.aeronomie.be/software/QDOAS/). The detailed configuration of spectral fitting are listed in Table 1. The fitting wavelength interval of $NO_2$, $O_4$, $SO_2$ and HCHO are 338-370 nm, 338-370 nm, 305-317.5 nm and 336.5-359 nm, respectively. Trace gases absorption cross sections of $NO_2$ at 220 K, and 298 K (Vandaele et al., 1998), $SO_2$ at 298 K (Vandaele et al., 2009), HCHO at 297 K (Meller and Moortgat, 2000), $O_3$ at 223 K and 243 K (Serdyuchenko et al., 2014), $O_4$ at 293 K (Thalman and Volkamer, 2013), BrO at 223 K (Fleischmann and Hartmann, 2004), the Ring spectrum, a

Frauenhofer reference spectrum and a low order polynomial are included in the DOAS fitting. The wavelength calibration was performed using a high resolution solar spectrum (Chance and Kurucz, 2010). Dark current spectrum and electronic offset spectrum were used to correct measurement spectra before the spectra analysis.

Table1.DOAS spectral fitting of $NO_2$, $O_4$, $SO_2$, and HCHO

| Parameter | Data source | Trace gases | | |
| --- | --- | --- | --- | --- |
| | | $NO_2$ & $O_4$ | $SO_2$ | HCHO |
| Wavelength range | | 338-370 nm | 305-317.5 nm | 336.5-359 nm |
| $O_4$ | Thalman and Volkamer (2013), 293K | √ | × | √ |
| $NO_2$ | Vandaele et al. (1998), 220K, 298K, $I_0$-correction ($10^{17}$ molec cm$^{-2}$) | √ | √(only 298K) | √(only 298K) |
| $SO_2$ | Vandaele et al. (2009), 298K | × | √ | × |
| HCHO | Meller and Moortgat (2000), 297K | √ | √ | √ |
| $O_3$ | Serdyuchenko et al. (2014), 223K, 243K, $I_0$-correction ($10^{20}$ molec cm$^{-2}$) | √ | √ | √ |
| BrO | Fleischmann et al. (2004), 223K | √ | √ | √ |
| Ring | Ring spectra calculated with QDOAS according to Chance and Spurr (1997) | √ | √ | √ |
| Polynomial degree | | 5 | 5 | 5 |
| Wavelength calibration | Based on a high resolution solar reference spectrum (SAO 2010 solar spectra) | | | |

The spectral analysis yields the measured Slant Column Densities (SCDs), the integrated trace gas concentration along the



light path through the atmosphere. For MAX-DOAS spectral analysis, the measured spectrum at 90° was selected as the Frauenhofer reference spectrum for the DOAS fitting of the measured spectra at other elevation angles in each scan sequence. So the generated results are the difference of the SCDs between the measured spectrum and that of the Fraunhofer reference spectrum, usually referred as Differential Slant Column Densities (DSCDs). Figure 2 shows a typical DOAS spectral fitting

of the measured spectrum collected at elevation of 15° at 10:13 local time (LT) on 7 June 2017. The retrieved DSCDs of $NO_2$ (Fig. 2a), $O_4$ (Fig. 2b), $SO_2$ (Fig. 2c), and HCHO (Fig. 2d) are $2.28 \times 10^{16}$, $1.90 \times 10^{43}$, $1.84 \times 10^{16}$, and $4.46 \times 10^{16}$ molec cm$^{-2}$, respectively. All these fittings displayed the evident absorption structures of the trace gases and fairly low residuals, which demonstrates the good performance of the spectral fitting. In this study, a threshold of residual $< 2.5 \times 10^{-3}$ are used to filter the unsatisfied fitting results of $NO_2$, $O_4$, and HCHO. Afterwards, the qualified DSCDs results remains 99.37%, 99.37%, and

99.79%, respectively. Considering the weak scattered sunlight signals and low signal-to-noise ratio around 300 nm, where the $SO_2$ has strongly structured absorption, the threshold of residual for $SO_2$ is set to $5.0 \times 10^{-3}$ and 70.05% of the fitting results are meet this criterion.

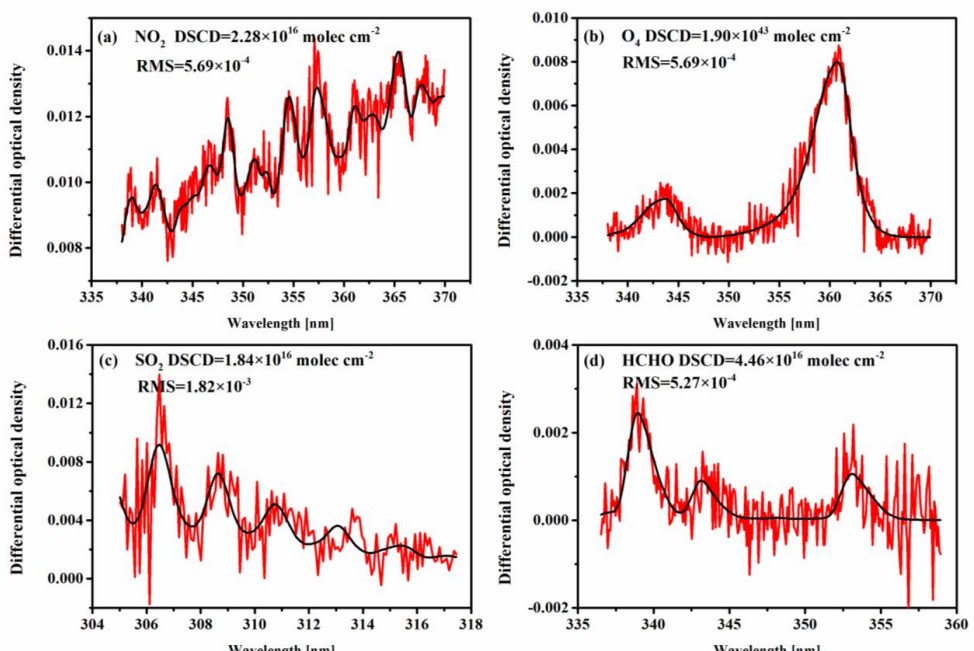

**Figure 2. Typical DOAS spectral fittings for (a) NO₂, (b) O₄, (c) SO₂, and (d) HCHO. The spectrum was collected at**

**elevation of 15° at 10:13 LT on 7 June 2017. The black curves show the reference absorption cross section scaled to measured atmospheric spectrum (red curves) by DOAS fitting.**

### 2.2.3 Retrieval of the trace gases VCDs and profiles

To obtain the tropospheric Vertical Column Density (VCD) of trace gas, the DSCDs has to convert using tropospheric





Differential Air Mass Factors (DAMFs) by Eq. (1) (Wagner et al., 2010):

$$VCD_{trop} = \frac{DSCDs}{DAMFs} = \frac{DSCDs(\alpha)}{AMF(\alpha) - AMF(90°)} \tag{1}$$

The Air Mass Factors (AMFs)calculation is approached by the so called geometric approximation method (Hönninger et al., 2004; Wagner et al., 2010), which is simple and convenient, simultaneously, is also validated by radiative transfer simulations (Solomon et al., 1987; Shaiganfar et al., 2011).

$$AMF(\alpha) = {1}/{sin(\alpha)} \tag{2}$$

Using Eq. (1) and Eq. (2), the tropospheric DAMF was estimated to be 2.86 and 1 for elevation angles of 15° and 30°, respectively. Previous ground-based MAX-DOAS studies shown that the most appropriate choice for the elevation angle could probably be 30° for the geometric approximation approach (Halla et al., 2011; Brinksma et al., 2008). Nevertheless, elevation 15° also works well for the conversion of DSCDs into VCDs in ship-based MAX-DOAS campaign since the last scattering point is generally above the trace gases layer for elevation angle 15° in the lower boundary layer over the sea (Schreier et al., 2015). In addition, due to the longer light path through the boundary layer, the observations at 15° elevation angle are more sensitive compared to 30°. Consequently, the VCDs of $NO_2$, $SO_2$, and HCHO were approached from DSCDs of 15° elevation angle by the geometric approximation method in this study.

To obtain the vertical distribution of trace gases, we used the HEIPRO algorithm (HEIdelberg PROfile, developed by IUP Heidelberg) for MAX-DOAS profile retrieval (Frieß et al., 2006; Frieß et al., 2011; Frieß et al., 2016). The HEIPRO retrieval algorithm is based on the Optimal Estimation Method (OEM, Rodgers, 2000) and coupled with the radiative transfer model SCIATRAN (Rozanov et al., 2005) as the forward model. Because the existence of aerosol has strong impacts on the scattered light path in the atmosphere, the retrieval algorithm takes into account the aerosol profile retrieval first and then adopts the retrieved aerosol scenario to profile the trance gases. In this study, an exponential decay a priori profile with a scale height of 1.0 km is used as the initial profile for both the aerosol and trace gases retrieval. The total aerosol optical depth and trace gases VCDs of such a priori profile is 0.2 and $7.27 \times 10^{15}$ molec cm$^{-2}$, respectively. The uncertainty of the aerosol and trace gases a priori profile is set to 100% and correlation length is set to 0.5 km. In the radiative transfer model, the parameters of single scattering albedo, asymmetry parameter and ground albedo are assumed to be 0.92, 0.68 and 0.06 for the marine environment. The retrieved profile of aerosol extinction and trace gases concentration has the resolution of a fixed grid of 200 m from sea surface to 3 km altitude. The criteria that the relative error of profile retrieval larger than 50% or degree of freedom of signal smaller than 1.0 are used to filter the profile results. Afterwards, about 1.1%, 23.4%, and 7.2% of all measurements were discard for $NO_2$ $SO_2$, and HCHO profile retrievals, respectively.

**2.3 OMI and OMPS satellite data**

The Ozone Monitoring Instrument (OMI) was launched on July 2004 on-board the NASA Aura satellite (Levelt et al., 2006).





It is an imaging spectrometer covering the wavelength range from 270 to 500 nm, which receives the light signal of scattered in the Earth's atmosphere and reflected by the Earth's surface. OMI aims to monitor ozone, $NO_2$, and other minor trace gases distribution with high spatial resolution (about $13 \times 24$ km$^2$) and daily global coverage. OMI is operated on a sun-synchronous orbit, and the overpass time is about 13:45 LT. However, OMI has suffered from a so-called "row anomaly" and lost several cross-track positions data (Boersma et al., 2011). In this study, we use USTC-OMI tropospheric $NO_2$ products (Liu et al., 2016; Su et al., 2017). To generate the USTC-OMI products, the $NO_2$ SCDs are retrieved from the OMI Level 1B VIS Global Radiances Data (OML1BRVG) based on the DOAS method. To convert into $NO_2$ VCDs more accurately, AMFs are calculated with the input of the localized $NO_2$ and atmospheric temperature and pressure profiles derived from WRF-Chem chemistry transport model simulations. The WRF-Chem chemistry transport model have using the National Centers for Environmental Prediction (NCEP) Final operational global analysis (FNL) meteorological data.

The Ozone Mapping and Profiler Suite (OMPS) instruments was launched on 28 October 2011 on board the Suomi National Polar-orbiting Partnership (Suomi-NPP) satellite. The OMPS Nadir Mapper (OMPS-NM) is one of three sensors of the OMPS suite instruments. It contains a UV spectrometer cover the wavelength range between 300 and 380 nm with a full width half maximum (FWHM) of 1 nm. It has a high spatial resolution of $50 \times 50$ km$^2$ and high time resolution of daily global coverage (Dittman et al., 2002; Seftor et al., 2014; González Abad et al., 2016). Its equator crossing time in the ascending node is 13:30 LT. In this study, the OMPS satellite observation data were used to retrieve the USTC-OMPS tropospheric $SO_2$ and HCHO products. Similar to the USTC-OMI tropospheric $NO_2$ VCDs, the $SO_2$ and HCHO VCDs are produced in two-steps approach too, i.e. first the SCDs of $SO_2$ and HCHO retrieval from the measured scattered sunlight spectra, and conversion to the $SO_2$ and HCHO VCDs by applying the calculated AMFs based the WRF-Chem chemistry transport model simulations results.

**2.4 Ozone Lidar**

During this campaign, an $O_3$ lidar was also on board co-located with the MAX-DOAS instrument, which was developed by Anhui Institute of Optics and Fine Mechanics (AIOFM) using DIfferential Absorption Lidar (DIAL) technology. The laser pulse of the lidar is at 316 nm, usually with the energy of about 90 mJ and a repetition frequency of 10 Hz. The laser beam is emitted with a divergence of 0.3 milliradian (mrad) and the receiving telescope with a field of view (FOV) of 0.5 mrad, resulting in an overlap height of approximately 300 m. The $O_3$ profiles in the lower troposphere were obtained using DIAL retrieval algorithms. The lidar observation has the high vertical resolution of 7.5 m and the temporal resolution about 12 min. In order to improve the signal to noise ratio, the retrieved vertical distribution $O_3$ concentrations were averaged in 100 m gridded. Additionally, the $O_3$ concentration profiles with relative errors above 20% were removed from the further discussion.




## 3    Results and discussion

### 3.1    Trace gases tropospheric VCDs

Based on the spectral analysis and the geometric AMF approach, we obtained the VCDs of different trace gases along the ship

cruise combined with the GPS received geo-position data. Figure 3 shows the spatial distributions of $NO_2$, $SO_2$ and HCHO

VCDs along the route over the ECS area. The missing data are due to the power failure and instrumental malfunction during

the campaign, as well as the measurements taken under bad weather conditions (e.g., heavy rain day). During the campaign,

the $NO_2$ VCDs varied from $1.00 \times 10^{15}$ molec $cm^{-2}$ to $5.52 \times 10^{16}$ molec $cm^{-2}$ with a mean value of $6.50 \times 10^{15}$ molec $cm^{-2}$. As

shown in Fig. 3(a), high level of the $NO_2$ VCDs, almost as three time as the average of the whole cruise, were observed at the

ship lanes of the south channel of the Yangtze River Estuary and the way to Lianxing port (located in Qingdong of Jiangsu

Province), as well as the busy port of Ningbo-Zhoushan area. The $SO_2$ VCDs are ranged from $1.00 \times 10^{15}$ molec $cm^{-2}$ to $1.77$

$\times 10^{16}$ molec $cm^{-2}$ with an average of $4.28 \times 10^{15}$ molec $cm^{-2}$. Fig. 3(b) shows the elevated $SO_2$ value (i.e. $> 8.63 \times 10^{15}$ molec

$cm^{-2}$, about the 95th percentile value, ~ 2.02 times of mean value) are appeared in the same places as $NO_2$, such as the ship

lanes closed to the Gongqing port, Ningbo-Zhousan port. For the HCHO, the averaged VCDs is $7.39 \times 10^{15}$ molec $cm^{-2}$ in

range of $1.02 \times 10^{15}$ molec $cm^{-2}$ to $3.16 \times 10^{16}$ molec $cm^{-2}$. As in Fig. 3(c), the enhanced HCHO columns were found in the

section of the cruise in Hangzhou Bay area, which is different with $NO_2$ and $SO_2$ spatial distribution. Moreover, high value of

HCHO VCDs $> 1.0 \times 10^{16}$ molec $cm^{-2}$ also dispersed over some hot spots as $NO_2$ and $SO_2$.





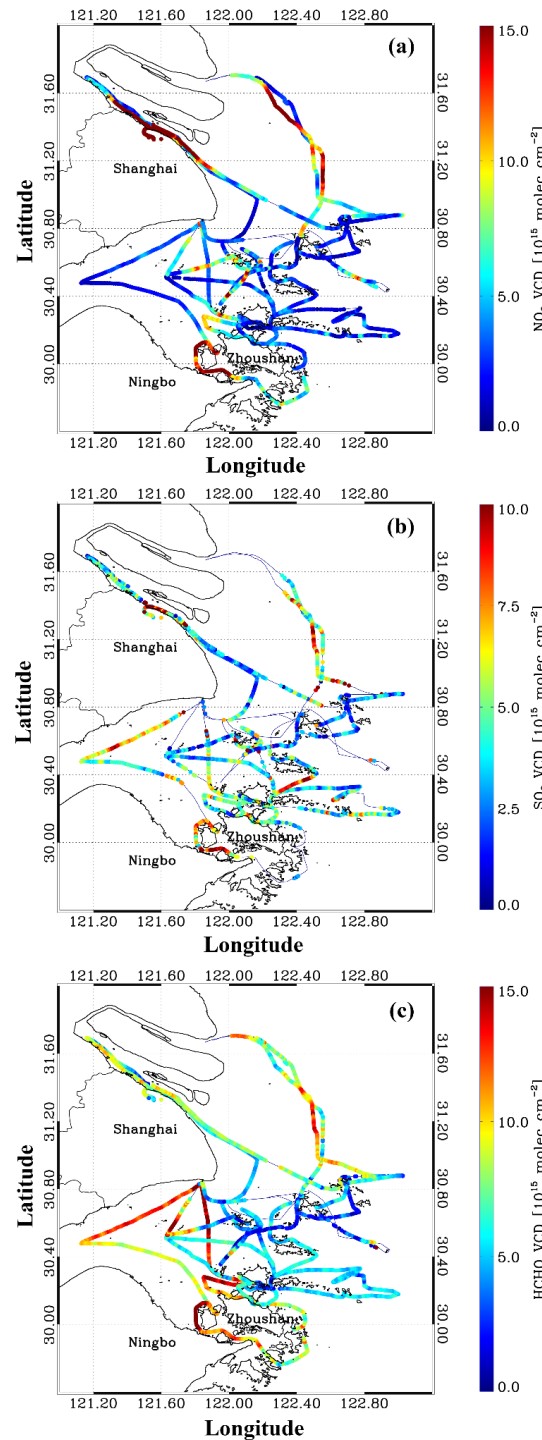

**Figure 3. The spatial distributions of the trace gases VCDs of (a) NO₂, (b) SO₂ and (c) HCHO along the cruise route of**

**the ship-based campaign in June 2017.**



The coastal waters of YRD region, including Jiangsu, Shanghai, and Zhejiang, is the busiest sea area of ECS, and the continental YRD region is also one of the most developed industrial city cluster of china or even the world. Therefore, previous studies found that the air qualities in coastal sea and inland area were affected by the ship-emitted pollutants under cruising and maneuvering conditions together with the continental anthropogenic pollutants (e.g. Zhao et al., 2013; Fu et al., 2014). In order to investigate impacts of ship emissions, we obtained the dependence of NO$_2$, SO$_2$ and HCHO VCDs on longitude in Fig. 4. It can be found that most of the peaks of trace gases are occurred at the geolocations of busy ports and ship lanes, whereas lower values are observed at remote oceanic area (Fan et al., 2016). The spatial distribution of NO$_2$, SO$_2$ and HCHO over sea areas are mainly dominant by the local emission source of ships, ports, and even the coastal factories.

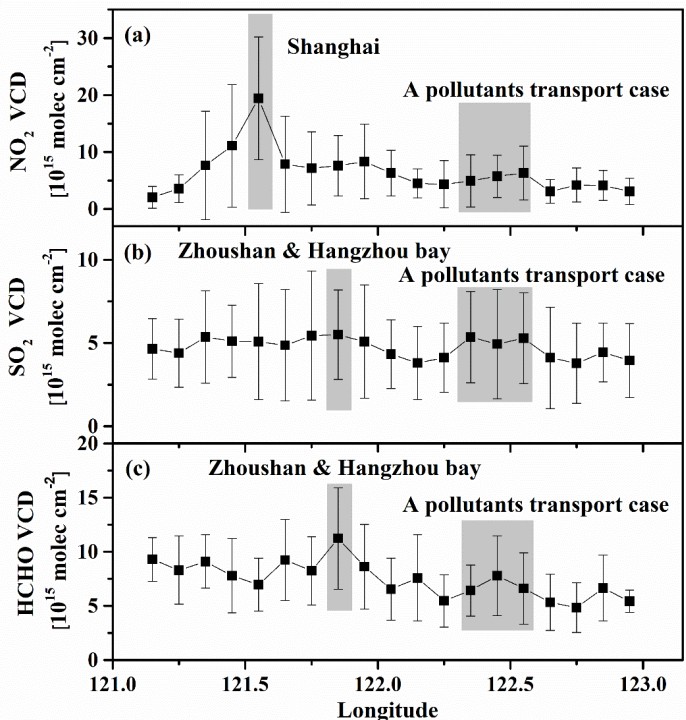

**Figure 4. The variations of trace gases tropospheric VCDs with longitude: (a) NO$_2$, (b) SO$_2$ and (c) HCHO.**

Besides, the spatial distribution of trace gases is also influenced significantly by the meteorological conditions especially wind speed and wind direction. Here, we calculated 24-h backward trajectories of 300 m altitude air masses by applying the HYSPLIT (Hybrid Single-Particle Lagrangian Integrated Trajectory) model, which is developed by the National Oceanic and Atmospheric Administration-Air Resource Laboratory (NOAA-ARL) (http://ready.arl.noaa.gov/HYSPLIT.php) (Stein et al., 2016). The Global Data Assimilation System (GDAS) meteorological data with a spatial resolution of 1°×1° and 24 vertical levels was used in the trajectory simulations process. Figure 5 displays the daily 24-h backward trajectories results for in four periods, which illustrated the origin of the air masses arrived at the endpoint (indicated by black triangle) at 04:00 UTC (12:00 LT).



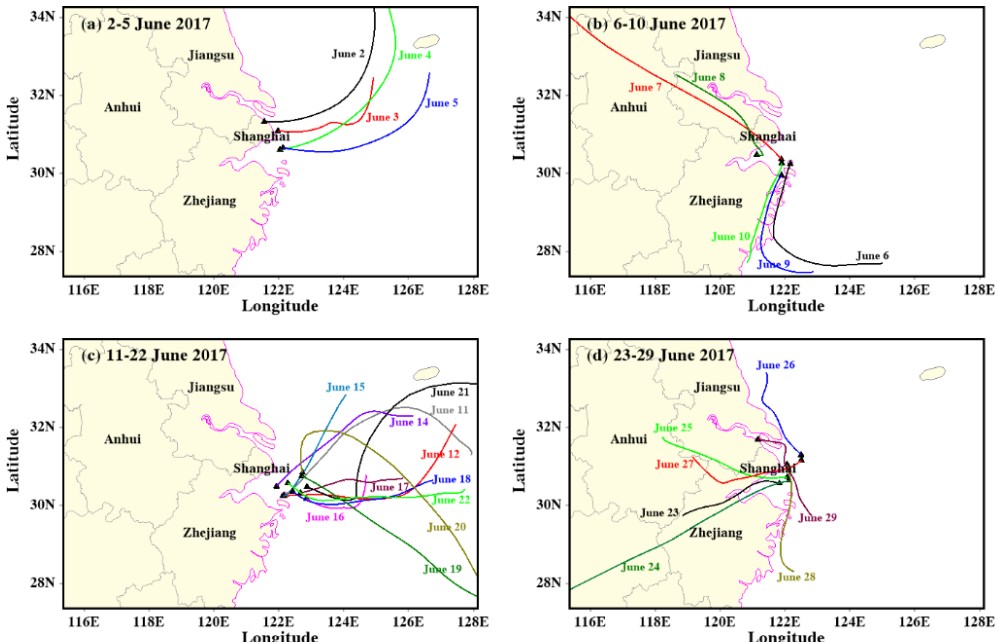


**Figure 5. Daily 24-h backward trajectories of air masses at the 300 m altitude for (a) 2 to 5 June, (b) 6 to 10 June, (c) 11 to 22 June and (d) 23 to 29 June, 2017.**

In Fig. 5(a) and (c), the air masses were originated from clean sea area of the duration from 2 to 5 and 11 to 22 June, 2017. It

suggested that the observed air pollutants were less impacted by the airflows transport, however, were mainly from the local

emission sources. For example, the high concentration of pollutants was reported on 2 and 3 June, during which the

measurements were implemented on busy ship lanes in the south channel of Yangtze River Estuary. In contrast, the trace gases

VCDs were much lower during the most days of 11 to 22 June, when the measurements were taken place over the clean sea

area. Figure 5(b) and (d) showed the air masses were came from inland area during 6 to 10 and 23 to 29 June, respectively. As

shown in Fig. 3, high values of $NO_2$, $SO_2$ and HCHO VCDs, the corresponding 95th percentile are $1.81 \times 10^{16}$, $1.05 \times 10^{16}$,

and $1.31 \times 10^{16}$ molec cm$^{-2}$ for these two days, have been found during the ship cruise from Shengsi islands to Lian Xiang port

and back to Huaniao islands on 26 and 27 June, 2017. The air mass was original from the coastal industrial zone on 26 June

and the city center of Shanghai on 27 June. This pollution episodes can be speculated mainly blame to pollutants transported

from inland cities and coastal areas combined with ship emissions in nearby waters.

**3.2     Comparison with OMI and OMPS satellite products**

In order to compare the ship-based MAX-DOAS and satellite data, we have to make them for comparable temporal and spatial

coverage. The ship-based MAX-DOAS measured VCDs are averaged for 13:00 to 14:00 according to the OMI and OMPS



instruments overpass time of about 13:45 LT and 13:30 LT. The satellite products are averaged within 10 km radius of the

center position of ship cruise between 13:00 and 14:00 LT considering the cruise speed around 8-15 km h$^{-1}$. Moreover, satellite

data with larger error (relative error >100%) and cloud impacts (could fraction > 0.5) were excluded from the inter-comparison.

Hence, there remains 14 days of observation for NO$_2$, SO$_2$ and HCHO VCDs comparison.

Figure 6(a) shows the time series of the NO$_2$ VCDs inter-comparison between ship-based MAX-DOAS measurements and

OMI satellite observations. These two data sets time agreed well with each other and have a high correlation coefficient (R) of

0.83 in Fig.6 (b). However, OMI satellite observations were higher than the ship-base MAX-DOAS results in some days,

which is different from the comparisons over continental areas where the satellite observation are usually much smaller than

ground-based data (Liu et al., 2016). The larger discrepancies on 8, 15 and 20 June were observed in the remote ocean area,

implying the possible larger uncertainties of the VCDs retrieval in such clear marine environment.

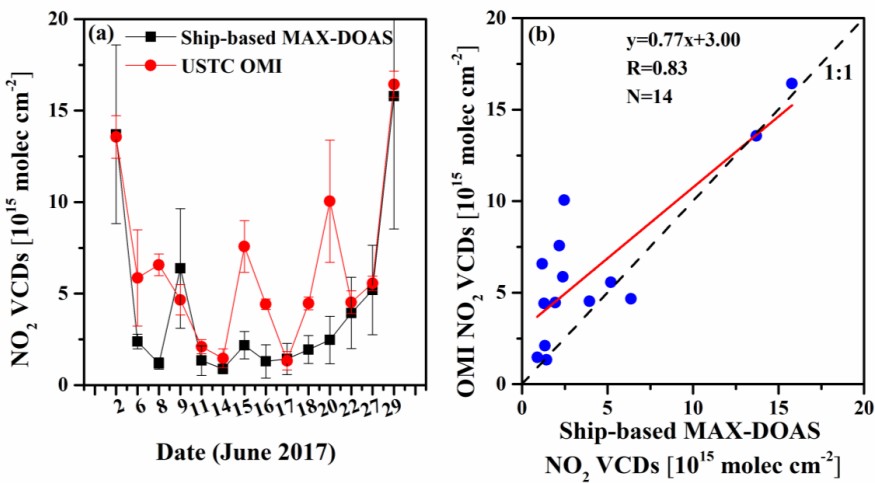

**Figure 6. Time series (a) and correlation analysis (b) of the tropospheric NO$_2$ VCDs measured by ship-based MAX-DOAS and OMI satellite during this campaign.**

For the inter-comparison with ship-based MAX-DOAS, the spaced products of SO$_2$ and HCHO VCDs were retrieved from

OMPS satellite. The time series of the SO$_2$ VCDs measured by ship-based MAX-DOAS and retrieved from OMPS satellite

observations were displayed in Fig. 7(a). These spaced and ship borne data exhibited similar temporal trends during the

campaign, showing a correlation coefficient (R) of 0.76 in Fig. 7(b). Figure 8(a) presents the time series of the HCHO VCDs

measured by ship-based MAX-DOAS together with the satellite data retrieved from OMPS observations, which also show a

good agreement with a correlation coefficient (R) of 0.69 in Fig. 8(b). Besides, we also found that the trace gases VCDs of

NO$_2$, SO$_2$ and HCHO from spaced observation by OMI and OMPS satellites are higher than ship-based MAX-DOAS

measurements in marine environment.



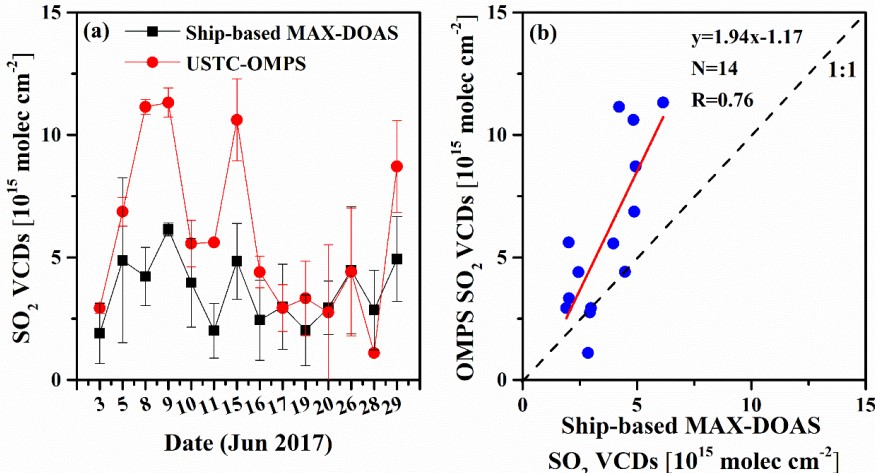

**Figure 7.** Time series (a) and correlation analysis (b) of the tropospheric SO₂ VCDs measured by ship-based MAX-DOAS and OMPS satellite during this campaign.

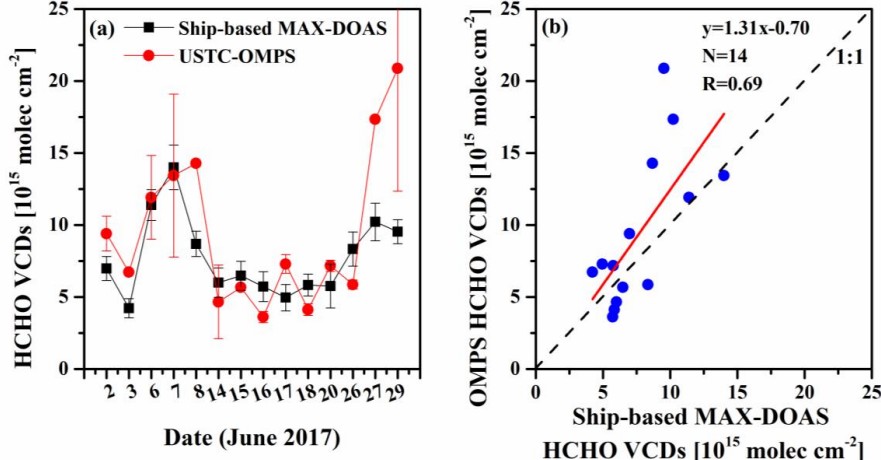

**Figure 8.** Time series (a) and correlation analysis (b) of the tropospheric HCHO VCDs measured by ship-based MAX-DOAS and OMPS satellite during this campaign.

To characterize the spatial distribution of tropospheric NO₂, SO₂ and HCHO VCDs, the monthly averaged tropospheric products of OMI satellite NO₂ and OPMPS SO₂, HCHO in June 2017 were demonstrated in Fig. 9. The satellite data was error (relative error >100%) and cloud (could fraction > 0.50) filtered and gridded in a high spatial resolution of 0.05°×0.05°. Due to the different emission sources and formation mechanisms, these three trace gases show the distinct features of spatial distributions. In Fig. 9(a), the hot spots of NO₂ distributions were distributed at the coastal of the Yangtze River at Shanghai city and Jiangsu Province, Ningbo-Zhoushan port and Shengsi islands. For the spatial distributions of SO₂ in Fig. 9(b), both



Qidong in Jiangsu province, northwest part of Shanghai city and the Hangzhou bay express relatively high values, and even

over some sea areas where are dense waterways and ship lanes. In addition, the main hot spots is located at urban area of

Shanghai city for HCHO spatial distributions in Fig. 9(c). In summary, all of the three trace gases have high values in some

polluted continental areas, e.g. Shanghai city center and northwest area, while hot spots over sea areas are mainly consistent

with the heavily vessels and ports emission areas.

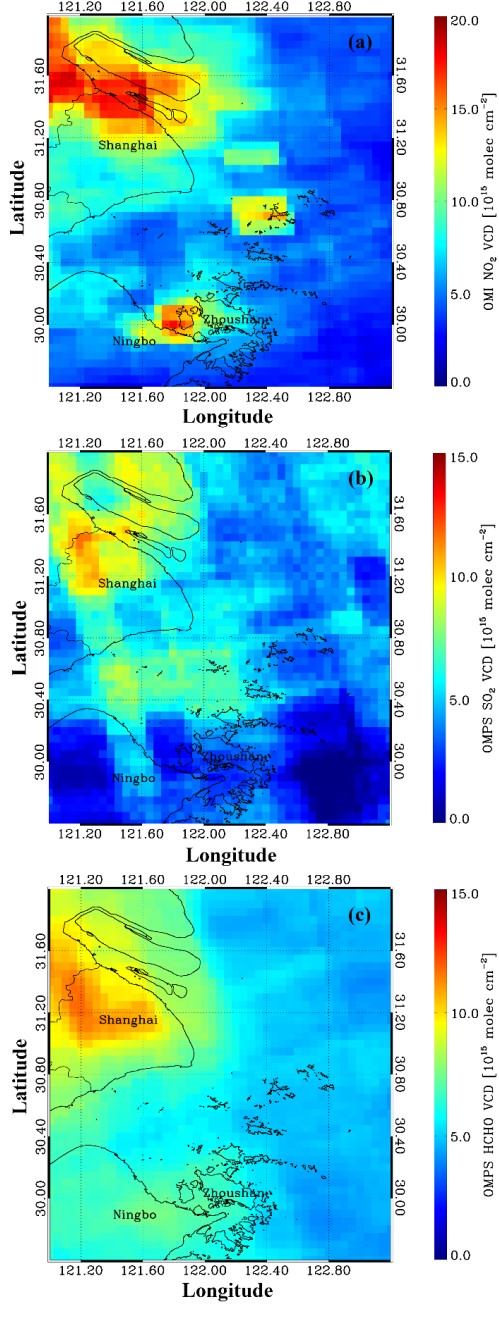





**Figure 9. The monthly averaged spatial distributions of the trace gases VCDs of (a) NO₂, (b) SO₂ and (c) HCHO of the**

**OMI and OMPS satellite observations in June, 2017.**

### 3.3 Tropospheric NO₂,SO₂ and HCHO profiles

In order to obtain vertical distribution of trace gases, we followed the method described in Sect. 2.4 to retrieve the vertical

profiles of $NO_2$, $SO_2$ and HCHO. Daily profiles results are available and three typical observation periods were presented for

different characteristic areas, as indicated in Fig. 10. The measurements during cycle 1 from 7 to 10 June were located at

Hangzhou Bay and Zhoushan islands areas. In cycle 2 from 16 to 19 June, the ship cruise were in the clean waters way far

from the coastlines. In cycle 3 of 26 to 29 June, the measurement carried on through a long path way, which passed through

clean to polluted areas over the waters of Shanghai, Qidong, and the areas of the Yangtze River Estuary.

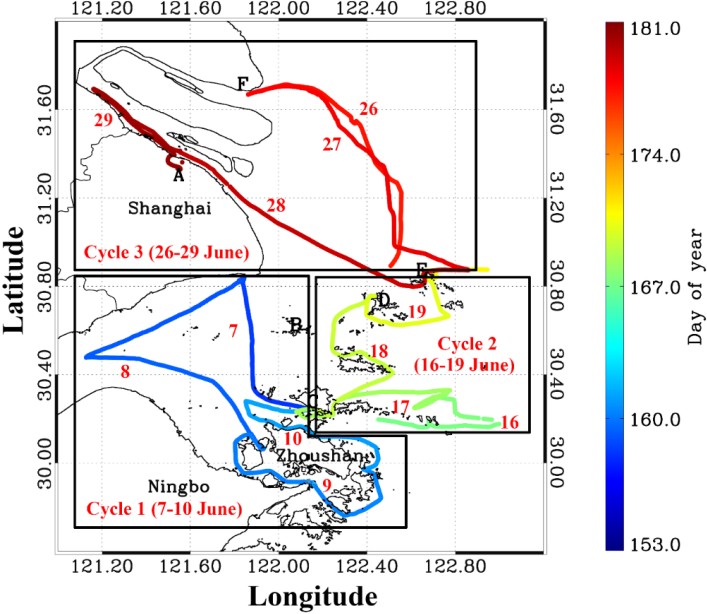

**Figure 10. Three typical observation periods in characteristic observation areas.**

     Figure 11(a)-(c) showed the diurnal variations of the vertical profiles of $NO_2$, $SO_2$ and HCHO concentrations during these

three cycles. It is obvious that the trace gases concentrations of $NO_2$, $SO_2$ and HCHO in cycle 2 were lower than that of the

others, which also can be confirmed in spatial distribution of trace gases VCDs in Fig. 3. By extracting the lowest 500 m grids

of the retrieved profiles, the observed concentrations of $NO_2$, $SO_2$, and HCHO were < 3 ppbv, < 3 ppbv and < 2 ppbv in such

clean marine boundary layer. It can be explained by the fact that the measurements of cycle 2 were performed in the relative

remote sea area, which is far away from the YRD continental region and less impacted by the inland emission sources, especial

under the favor meteorological condition of clean air masses from the remote ocean (in Fig. 5(c)). However, the higher levels

of the trace gases were found during cycle 1 and 3. For example, the concentrations of $NO_2$, $SO_2$ and HCHO in the marine





boundary layer all increased to high values on 9 and 29 June.

To track the cruise on 9 June, the ship was passing through the channel between Ningbo and Zhoushan islands, where is the Ningbo-Zhoushan Port, the world top 1st port ranking by container throughput. The Ningbo-Zhoushan Port has been reported to account for about one-third of the national port-level emissions inventories in China (Fu et al., 2017). The hourly variation

of pollutants emissions behaved to reach peaks during 9:00~14:00 and varied with vessel types (Yin et al., 2017). The backward trajectory on 9 June shows that air mass was originated from the coastal area. It is inferred that this pollution episode was mainly attributed to the ship emission from coastal and ocean going vessels, as well as the cargo handling equipment in the ports areas. On 29 June, the observation was performed along the Yangtze River toward upstream until 121.15°E and then back to Gongqing port of shanghai. This waterway is the only channel to the upstream of the Yangtze River and consequently

dense with the inland ships. Moreover, Taicang Port and some industrial zones were distributed along the coastline areas. Therefore, industrial factories and ship emissions, as well as the transports from inland city, lead together to this elevated pollutants levels.

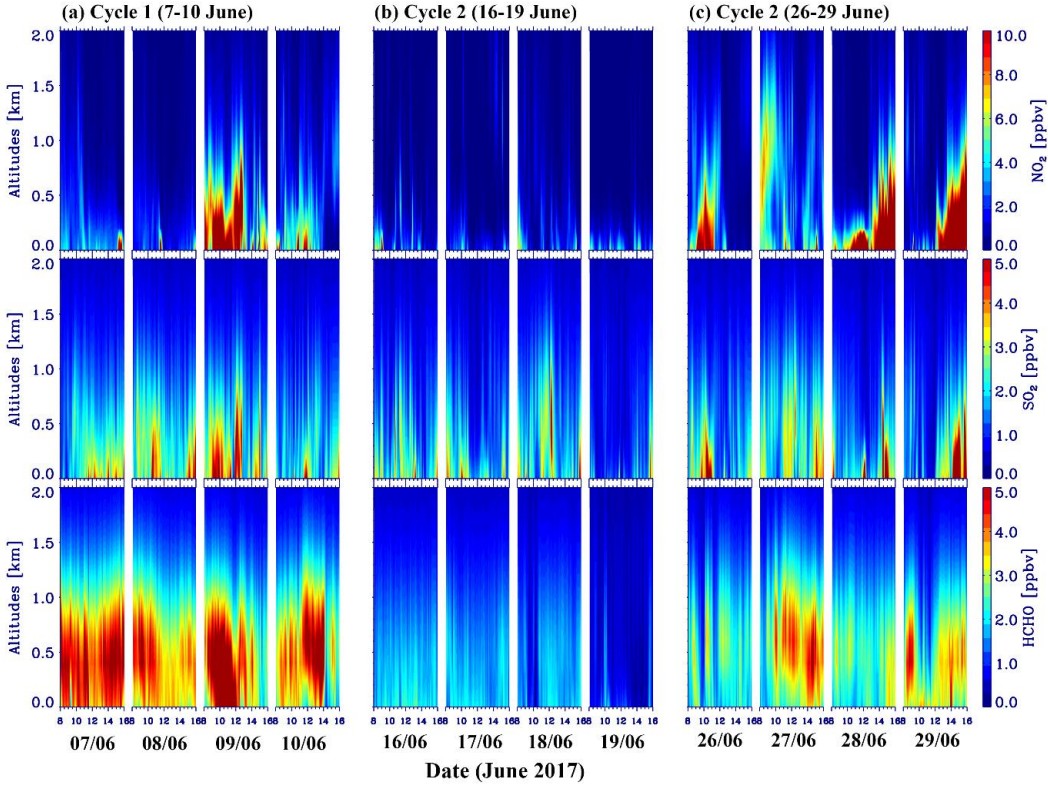

**Figure 11. Vertical profiles of NO₂, SO₂ and HCHO concentrations during the three typical observation periods: (a)**

**measurements taken at Hangzhou bay and Zhoushan islands on 7 to 10 June, (b) measurements carried on a relative clean area on 16 to 19 June, and (c) measurements implemented on the areas of the Yangtze River Estuary on 26 to 29**

**June.**

In addition, the vertical distributions of $NO_2$, $SO_2$ and HCHO in marine boundary layer have unique features. The high $NO_2$

concentrations were observed closed to the sea surface and decreased with height in vertical. For the layer below 500 m, lowest

and highest $NO_2$ concentrations were found < 3 ppbv in cycle 2 and > 10 ppbv during cycle 1 and 3, respectively. Almost all

the measured $NO_2$ concentrations in the marine boundary layer during this campaign are larger than the background value over

the western Pacific and Indian Ocean (< 0.2 ppbv) (Takashima et al., 2012) and over the South China and Sulu Sea (< 30 pptv)

(Peters et al., 2012). Due to the sulfur-containing marine fuels, ship emissions are the primary source of $SO_2$ over the seas.

Therefore, the intermittent enhanced $SO_2$ signals were detected during the whole cruise as shown in Fig. 11, even for the

relative clean area in cycle 2 where the $SO_2$ concentrations exceeded 3 ppbv sometimes. It implies that the frequently observed

$SO_2$ pulses are the emissions from the kinds of vessels in the vicinity or even from the cruise ship itself.

As distinguished from patterns of $NO_2$ and $SO_2$ vertical profiles, the highest HCHO concentrations are located at the elevated

altitudes (about 500 m) during the days of cycle 1 and 3, since there is no HCHO sources from the sea surface. The similar

phenomenon was also reported in the study over remote western Pacific Ocean, where the highest concentrations of HCHO

occurred at the altitudes of 400 m (Peters et al., 2012). Furthermore, extremely high HCHO concentrations of > 5 ppbv

appeared during the ship cruised along coastal and busy ports areas in cycle 1 and 3, while the low concentration about 1.2

ppbv were measured in cycle 2. However, the observed lowest levels HCHO in marine boundary layer of ECS area were almost

equal to the highest value (~ 1.1 ppbv) measured at remote western Pacific Ocean (Peters et al., 2012). The behavior of $NO_2$,

$SO_2$ and HCHO concentrations highlighted the obvious shipping emissions along the ship lanes and close the busy ports, and

further significant impacts on the regional air quality over the ECS areas.

### 3.4 Ozone formation

Figure 12 presents the on-board DIAL observed vertical distributions of ozone concentrations from 300 m up to 2 km Above

Sea Level (ASL) during the campaign. Except the absence on 13 and 28 June due to the power failure, there were 26 days

measurement results. It is found that the ozone concentrations in the marine environment showed a characteristic vertical

structure that the $O_3$ concentrations increased with the altitude from 300 m to 1.0 km ASL, and high values > 100 ppbv were

mostly distributed at altitudes higher than 1 km ASL. However, the high ozone concentrations were detected from 300 m and

spread to 1.4 km on 7 and 8 June. For the diurnal patterns, the $O_3$ concentrations usually began to increase at morning with the

enhancing solar radiation, and accumulated to arrive the daily peak in the afternoon, then declined with the decreases of sun

illumination. The similar diurnal variations were also reported in previous studies over continental area, e.g. the measurements

in vicinal Hangzhou, Zhejiang Province (Su et al., 2017). It indicated the daytime intense photochemical processes in the



marine boundary layer.

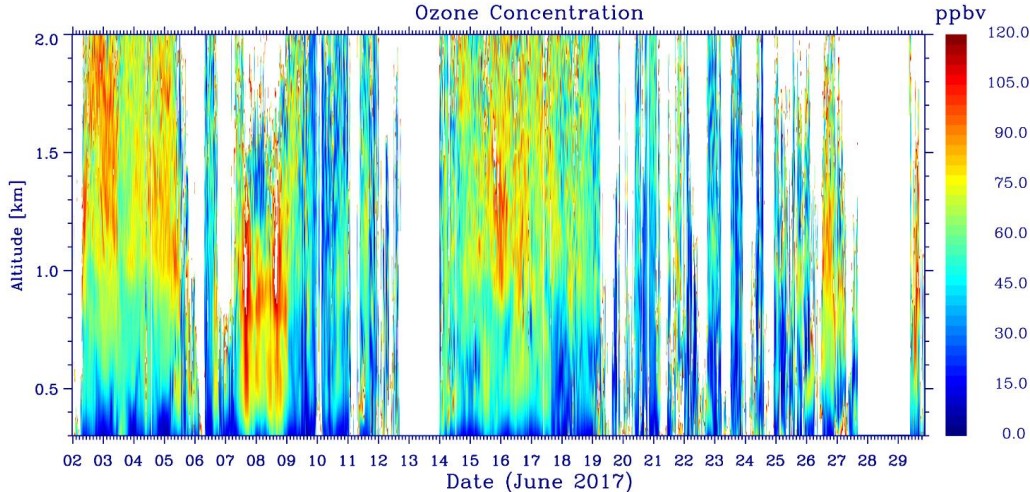

**Figure 12. Time series of the vertical profiles of the O$_3$ concentrations measured by the ozone lidar in June 2017.**

In order to investigate the formation and consumption processes of ozone, we integrated the vertical profiles of O$_3$ by lidar and NO$_2$, HCHO profiles from MAX-DOAS together. We averaged the daily profiles of O$_3$ and ratio of HCHO/NO$_2$ during the intense photochemical periods between 09:00 and 15:00 LT. Figure 13 shows the vertical-resolved comparisons between O$_3$ and HCHO/NO$_2$ ratio profiles on 7 to 10 June. Referred to Fig. 11, the NO$_2$ concentrations were higher in 9 and 10 but lower in 7 and 8, while the HCHO concentrations kept in high levels during the whole cycle 1 of 7 to 10, June. Accordingly, the ratios of HCHO to NO$_2$ on 7 and 8 were higher than that of 9 and 10, June. Meanwhile, the O$_3$ concentrations ranged at different altitudes from 70 to 100 ppbv in the marine boundary layer on 7 and 8, however, were below 60 ppbv on 9 and 10, June. It can be inferred that the high O$_3$ episodes on 7 and 8 June were controlled by the NOx-regime of ozone formation, because the O$_3$ concentration dropped significantly with the increases of NO$_x$ concentration and simultaneous decreases of HCHO/NO$_2$ ratio.

As shown in Fig. 5(b), air masses on 7 and 8 June originated from northwest inland area, however, from southwest coastal area on 9 and 10 June. The air masses transportation from different regions may also contribute to this ozone pollution episode. Furthermore, it can be concluded that the high O$_3$ concentrations exceeding 60 ppbv can be expected while the ratios of HCHO to NO$_2$ larger than 1.5 and vice versa during this case.





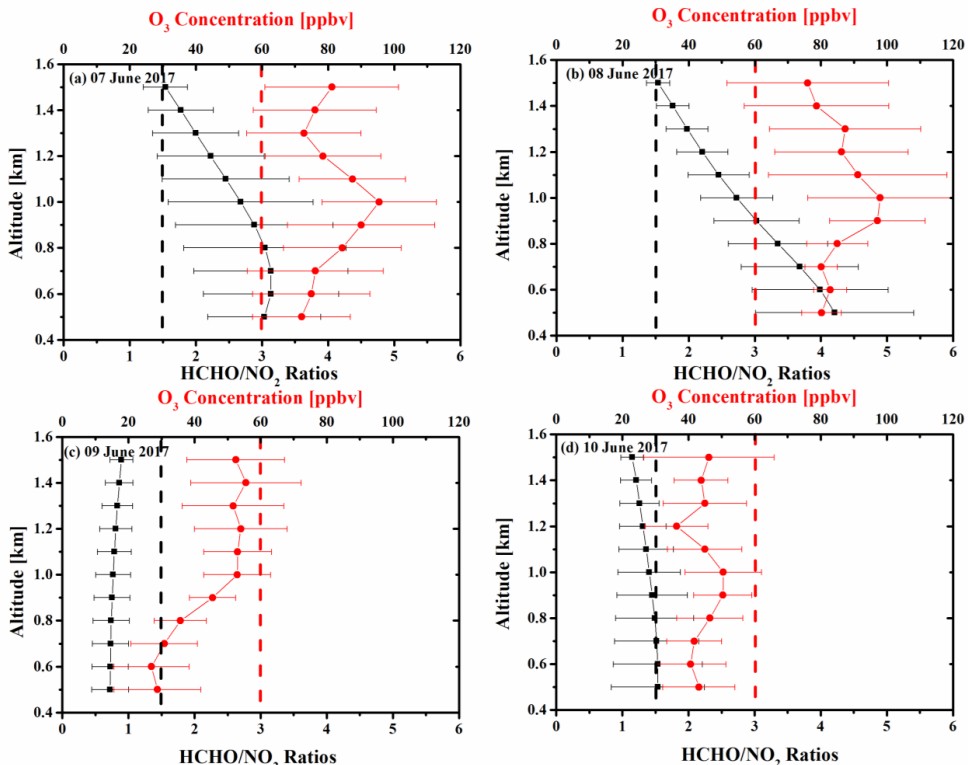

390 **Figure 13. The daily averaged vertical profiles of O₃ concentrations and HCHO/NO₂ ratios at different altitudes between 09:00 and 15:00 on 7 to 10 June, 2017.**

## 4 Summary and conclusions

In this paper, ship-based MAX-DOAS and ozone lidar measurements were performed in the YRD region over the ECS area

395 from 2 to 29 June 2017. During this campaign, the measured VCDs of $NO_2$, $SO_2$ and HCHO were firstly reported to be $6.50 \times 10^{15}$, $4.28 \times 10^{15}$ and $7.39 \times 10^{15}$ molec $cm^{-2}$, respectively for ECS area. In order to validate the ship-based MAX-DOAS measurements, tropospheric $NO_2$, $SO_2$ and HCHO VCDs were compared with the satellite observations. Both $NO_2$, $SO_2$ and HCHO showed good agreements between MAX-DOAS results and satellite products with a correlation coefficient R of 0.83, 0.76 and 0.69, respectively. Furthermore, the spatial distribution of trace gases along the ship cruise demonstrated that the

400 enhanced pollutions of trace gases are usually related to the emissions from vessels in vicinal waterways and busy ports area. In general, the levels of trace gases decreased with the distance from the coastlines, whereas the exceptional case that high values observed on 26 and 27 June at relative remote sea areas was mainly owing to the transport process from continental area with the favor of meteorological conditions.

The daily vertical profiles $NO_2$, $SO_2$, and HCHO were obtained by the retrieval from MAX-DOAS measurements by HEIPRO

algorithm. The trace gases concentrations in the bottom of marine boundary layer are $< 3$, $< 3$, and $< 2$ ppbv of $NO_2$, $SO_2$, and

HCHO respectively over the relative clean areas far from offshore of the YRD region. However, we also found elevated $SO_2$

concentration frequently during the cruise, which is blamed to the ships emissions nearby. Combined the ratio of HCHO/$NO_2$

profiles from ship-based MAX-DOAS with $O_3$ vertical profiles from the ozone lidar, the typical $O_3$ formation were identified

related to the increases of the $NO_2$ concentration and relative lower HCHO/$NO_2$ ratios. This study highlighted the strong

impacts of shipping emissions on the air quality in marine boundary layer of ECS areas, which need to be regulated urgently

in the coming future, especially for the YRD region where the world top 2 ports are located.

**Acknowledgements**

This research was supported by grants from National Key Research and Development Program of China (2016YFC0203302,

2017YFC0210002), National Natural Science Foundation of China (41722501, 91544212, 51778596, 41575021, 41775113).

We acknowledge the NOAA Air Resources Laboratory (ARL) for making the HYSPLIT transport and dispersion model

available on the Internet (http://ready.arl.noaa.gov/). We would like also to thank Fudan University to organize the ship-based

campaign and Hefei Institute of Physical Science, Chinese Academy of Sciences for the technical support of lidar measurement.

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
