# Peer review of "Tropospheric NO2, SO2, and HCHO over the East China Sea, using ship-based MAX-DOAS observations and comparison with OMI and OMPS satellites data"

_Atmospheric Chemistry and Physics, 2018_

## Referee Comment (RC1) · M. Wenig (Referee) · 21 Aug 2018

The manuscript "Tropospheric NO2, SO2, and HCHO over the East China Sea, using ship-based MAX-DOAS observations and comparison with OMI and OMPS satellites data" by Wei Tan et al. describes a nice combination of different measurement techniques to produce a very interesting data set of NO2, SO2 and HCHO VCDs to validate satellite retrievals. The individual measurement techniques are well established, so the paper is not presenting a novel idea, but rather a thoroughly executed concept to produce a very valuable satellite validation data set. The conclusions are clear, but could

go into more detail. Can the results be used to improve satellite retrievals over similar marine areas? Does this study prove that ship emissions in this area are stronger than previously assumed? The scientific methods are described in detail, but I'm still a little confused about the MAX-DOAS retrieval. You describe two types of retrievals, a VCD retrieval based on a geometric AMF from one elevation angle (plus zenith for reference) and a more precise one based on optimal estimation using the full scan of 7 angles. If you have the time to perform full scans, why do you include the first VCD retrieval? Did you compare the two results, the VCDs from single elevation angles and VCDs derived from the profiles?

The MAX-DOAS retrieval provides aerosol profiles as a first step to derive trace gas profiles. Did you look at the aerosol profiles as well? Do they justify using the trace gas VCD retrieval based on the geometric AMF which doesn't account for aerosols? For the retrieval you calculate dAMFs anyway, why do you use a geometric approximation then? This part could be described in more detail, in order to avoid confusion about the retrieval process. Using only 2 elevations is typically used in a mobile setup when the concentrations change quickly, e.g. in traffic, and a full scan typically in a fixed setup where you have enough time under stable conditions to scan several elevations angles. Maybe you switch between the two scanning modes depending on what scenario is more appropriate? Your spectrometer covers the range 300-460nm, why don't you use the 400-460nm wavelength range for the NO2 retrieval, where NO2 has pronounced absorption structures and is less influenced by O3 absorption? The presentation of the results is well structured and the title and abstract reflect the contents of the paper perfectly. The language is okay but there are a few sentences that could need some improvements as follows:

page 2, line 56 "quantify kinds of the atmospheric trace gases" -> "quantify different kinds of atmospheric trace gases" (that sounds more fluent to me, but it's just a suggestion)

page 2 line57 "Based on the DOAS principle, the quantitative of the trace gases was

acquired from the narrow band absorption structures of the different trace gases , which were separated from the broad and parts caused primarily by the atmospheric scattering and their broad band absorption (Platt and Stutz, 2008)" -> "The DOAS principle makes use of the fact that narrow trace gas absorption structures can be separated from broad band absorption and atmospheric scattering (Platt and Stutz, 2008)" (or something like that)

page 2 line 59 "The named Multi-AXis-Differential Optical Absorption Spectroscopy (MAX-DOAS) instrument is designed . . ."->"The Multi-AXis-Differential Optical Absorption Spectroscopy (MAX-DOAS) instrument is designed . . ."

page 3 line 69 ". . .trace gases concentrations. . ." -> ". . .trace gas concentrations. . ."

page 4 l98 "The cruise of ship-based observation" -> maybe just "The measurement cruise"?

page 4 line99 "The ship-based measurements campaign" -> "The ship-based measurement campaign"

page 12 line254 ". . .impacted by the airflows transport. . ." -> ". . .impacted by airflow patterns. . ."

page 12 line258 ". . .showed the air masses were came from inland area . . ." -> ". . .showed the air masses coming from inland areas . . ."

line272 "These two data sets time agreed well. . ." -> "These two data sets time agree well . . ."

All in all the authors present valuable results in a well structured paper and I recommend publication in ACP after the above mentioned comments have been addressed.

---

## Referee Comment (RC2) · Anonymous Referee #3 · 21 Aug 2018

The authors of this article present an effort to investigate the trace gases over the East China Sea with ship-based optical measurements and satellite observations. Considering that few ground-based observations are available over the seas near eastern China by now, these results can provide an important reference for the community. The novelty and expression of the paper should be improved besides description of the observation data. Here are some specific comments: 1. What's the purpose of comparison between ground measurements and satellite retrieval hereïij§In line 396 of section 4, the authors said "In order to validate the ship-based MAX-DOAS measurements...",

[Figure]

**[ACPD](ACPD)**
however, satellite retrieval of trace gases have considerable uncertainties. 2. Line 267, 10 km radius of location of ship-based measurements is selected to match with satellite pixel, but the satellite pixel especially the OMPS is much larger than this scope, how realize it? 3. The current data analysis did not well support the authors' conclusion robustly. Daily satellite observation can provide regional view of the distribution of gaseous pollutants, why the authors only show monthly data? How did the daily satellite data compare with of daily values of in the track of ship-based measurements? To reveal the transport and air pollution over sea, typical daily case is suggested. 4. It is important for the authors to clarify and emphasize what's new in their work and what's their new finding?

---

## Author Comment (AC1) · 29 Sep 2018

We truly grateful for the reviewers' positive assessments of our manuscript and the helpful suggestions. We have revised the manuscript carefully according to the reviewers' comments. Point-to-point responses are given below, the original comments are black in color, while our responses are in blue. The line number are referred to the revised manuscript.

**General comments**

(1) Can the results be used to improve satellite retrievals over similar marine areas?

R: In general, the ship-based measurements are very helpful for the improvements of satellite retrievals over sea areas. For example, the ground-based data are rarely reported for the open oceanic areas, so there are lack of the validations for satellite products. Moreover, the ship-based MAX-DOAS results can provide the relatively actual vertical distributions of atmospheric aerosol and trace gases, which can be introduced into the retrieval scheme as the input parameters of forward radiative transfer model to improve the accuracy of satellite products over sea areas.

However, the spatial coverages of ship-based measurements are limited to the cruise track, which is insufficient compared to the satellites spatial resolution in this study. So it is difficult to directly assimilate the measured profiles by ship-based measurements to recalculate the AMF using for satellite SCD conversion to VCD in large scale. Nevertheless, we used the simulated profiles from WRF-chem model with a spatial resolution of $20\,km \times 20\,km$ to recalculate the AMF for satellite retrieval over the whole measurement areas. Consequently, the USTC's satellite products were confirmed more accurate than NASA's products in previous studies (Hong et al., 2018; Liu et al., 2016; Xing et al., 2017).

(2) Does this study prove that ship emissions in this area are stronger than previously assumed?

R: Generally, there could be two approaches to quantify the ship emissions using the ship-based MAX-DOAS measurements. The first idea is the measuring of pollutants emission of individual ship and obtaining the data of ship numbers and activities for

summation of the total emissions. In this study, the MAX-DOAS instrument has not been used to track the ship plume. So it's impossible to quantify the ship emission of single ship, suggesting that this approach is not suitable herein. Alternatively, the ship-based MAX-DOAS measurement can be used to encircle a sea area to quantify the emissions of this encircled area like the ground-based mobile vehicle-borne application. Then, the ship emission can be further determined by the difference between the fluxes entering and leaving the areas (Wang et al., 2012).

So we tried to quantify the ship emissions using the emission flux calculation method described in previous mobile DOAS studies (Shaiganfar et al., 2017; Shaiganfar et al., 2011; Wu et al., 2013). According the cruise track in Fig. 1(b), it is hard to find an encircled area during the campaign. So we selected the measurements on 27 June as a case study. As shown in Figure R1, we first assumed an encircled area, indicated with the black rectangle, along the ship track for emission estimation. The downwind boundary line of the area (marked by the letter c) were observed by ship-based measurements.

[Figure]

*Figure R1. A sketch map of emission estimation, the colored line is the cruise track of ship-based measurements on 27June 2017, while the black rectangle indicates an assumed encircled area with one boundary line (marked by the letter c) were observed by ship-based measurements.*

The flux transported through this boundary can be can be achieved though the formula:

$Flux \approx \sum VCD_i \cdot \omega_i \cdot v_i \cdot \Delta t_i$, where $\omega$ is wind speed, $v$ is ship speed. If the fluxes passed into the area from other boundary lines were not considered, the *Flux* can be used to represent the emissions of this area but with significant overestimation.

Under this simplification and assumption, we have chosen a period of 1-hour ship-based measurements, corresponded to the travel distance of 14.4 km along the boundary (c). With an averaged wind speed of 2 m/s on 27 June, the emission results of $NO_2$ and $SO_2$ were estimated to be $9.19 \times 10^{24}$ molec/s and $2.03 \times 10^{24}$ molec/s for this assumed area (103.68 $km^2$), respectively.

Compared to other studies, it can be found that the estimated emission of $NO_2$ (214.31 tonnes/yr/$km^2$) and $SO_2$ (65.86 tonnes/yr/$km^2$) are much higher than previous reported data for this sea area (Fan et al., 2016; Fu et al., 2017). Fu et al. (2017) have reported that the annual emission intensity of $NO_x$ and $SO_2$ in the Yangtze River Delta (YRD) area (where the MAX-DOAS measurements were performed) were 3.76 and 2.78 Ton/$km^2$. The statistical average intensities of NOx and $SO_2$ were 17 and 7.1 tonnes/yr/$km^2$ at the intersection hub of the coastal shipping lanes and the Yangtze River Channel, however, the highest regional emission intensities of NOx and $SO_2$ can increase to $1.0 \times 10^4$ and $1.3 \times 10^4$ tonnes/yr/$km^2$, respectively (Fan et al., 2016).

Therefore, the observed ship emission by MAX-DOAS are relatively reasonable and comparable. The overestimations can be mainly explained by:

(1) In this simplified pattern, the fluxes transported into the area have not been taken into account, which lead to the overestimations to some extent.

(2) The ship emissions are concentrated in the ship lane and port areas, showing the obviously different emission intensities in spatial pattern. Due to the different areas of these researches, the comparison may result larger discrepancies.

(3) Another possible reason could be that the different methodology of emission estimation also yields the systematic deviation.

According to the discussion above, it can prove that the MAX-DOAS method is applicable for the ship emission estimation in practical. However, the operations of measurements and cruise need to be designed more delicate, if the ship-based MAX-

DOAS is aiming to quantify the ship emission over oceanic area. For example, the cruise of ship-based measurements can be planned to encircle a sea area, and the telescope has been pointed to the ship smokestack for the scanning of emission plume.

(3) The scientific methods are described in detail, but I'm still a little confused about the MAX-DOAS retrieval. You describe two types of retrievals, a VCD retrieval based on a geometric AMF from one elevation angle (plus zenith for reference) and a more precise one based on optimal estimation using the full scan of 7 angles. If you have the time to perform full scans, why do you include the first VCD retrieval? Did you compare the two results, the VCDs from single elevation angles and VCDs derived from the profiles?

R: The geometric AMF approach is a simple and quick method to obtain the VCDs, which is also verified by optimal estimation method. Previous study shows that the results by these two methods were highly consistent, especially in marine environment, where the atmospheric boundary layer are usually very low (Schreier et al., 2015). As shown in Figure R2, we have compared the VCDs obtained by these two different ways. VCDs of $NO_2$, $SO_2$, and HCHO showed a good agreement between these two methods with a correlation coefficient R of 0.94, 0.76, and 0.93, respectively.

[Figure]

*Figure R2. Comparison of (a) $NO_2$, (b) $SO_2$ and (c) HCHO VCDs obtained by geometric AMF approach and derived from the profiles.*

(4) The MAX-DOAS retrieval provides aerosol profiles as a first step to derive trace gas profiles. Did you look at the aerosol profiles as well? Do they justify using the trace gas VCD retrieval based on the geometric AMF which doesn't account for aerosols?

For the retrieval you calculate dAMFs anyway, why do you use a geometric approximation then?

R: As described in Section 2.2.3 of the manuscript, we used the HEIPRO algorithm for MAX-DOAS profile retrieval during this campaign, which is based on the Optimal Estimation Method (OEM, (Rodgers, 2000)). The retrieval algorithm takes into account the aerosol profile retrieval firstly and then adopts the retrieved aerosol scenario to profile the trace gases.

Since we focused on the trace gases in this paper, no aerosol retrieval results were presented. However, aerosol profile retrieval is the precondition of trace gases profile. We always take into account the aerosol profiles first. For example, Figure R3 shows the aerosol profiles corresponding to the three cycles in Figure 12 of the manuscript.

The reason of using geometric AMF to obtain trance gases VCDs and the verification were also discussed above. Please refer to the responses to Q(3).

[Figure]

*Figure R3. Vertical profiles of aerosol extinction during the three typical observation periods same as Fig. 12 in manuscript.*

(5) Using only 2 elevations is typically used in a mobile setup when the concentrations change quickly, e.g. in traffic, and a full scan typically in a fixed setup where you have enough time under stable conditions to scan several elevations angles. Maybe you switch between the two scanning modes depending on what scenario is more appropriate?

R: The typical cruising speed of the measurements is about 5 m/s, and one full scanning sequence of 7 elevation angles takes about 4 min. So the travel distance during one sequence is about 1.2 km. Considering the relatively short time period and distance, it

can be treated as a stable condition. So we just used this multi-elevation angles scanning mode in the campaign.

(6) Your spectrometer covers the range 300-460 nm, why don't you use the 400-460 nm wavelength range for the NO2 retrieval, where NO2 has pronounced absorption structures and is less influenced by O3 absorption?

R: As we all know, the pronounced $NO_2$ absorption structures cover large wavelength range, so the $NO_2$ DSCD can easily be retrieved in the wavelength interval of 338-370 nm too. In addition, the $O_4$ has strong absorption peaks at 360.8 nm, which can be covered by the fitting window of 338-370 nm. $NO_2$ and $O_4$ fitting in same wavelength range is convenient for the further $NO_2$ profile retrieval, which is based on the aerosol information retrieved by $O_4$ DSCD. Besides, the range 338-370 nm is also wildly employed for $O_4$ and $NO_2$ retrieval together in previous DOAS studies (Hong et al., 2018; Johansson et al., 2008; Roscoe et al., 2010; Wu et al., 2018).

Here, we also analyzed the $NO_2$ DSCDs in the fitting window of 400-460 nm. Figure R4 presents the comparison of the $NO_2$ DSCDs retrieved in 338-370 nm and 400-460 nm wavelength range. The results of two different fitting windows agreed very well with each other, showing a high correlation coefficient R of 0.986 and the slope is 0.98.

[Figure]

*Figure R4. Comparison of NO₂ DSCDs retrieved using 338-370 nm and 400-460 nm wavelength ranges.*

**Technique corrections**

① page 2, line 56 "quantify kinds of the atmospheric trace gases" -> "quantify different kinds of atmospheric trace gases" (that sounds more fluent to me, but it's just a suggestion)

② page 2 line57 "Based on the DOAS principle, the quantitative of the trace gases was acquired from the narrow band absorption structures of the different trace gases, which were separated from the broad and parts caused primarily by the atmospheric scattering and their broad band absorption (Platt and Stutz, 2008)" -> "The DOAS principle makes use of the fact that narrow trace gas absorption structures can be separated from broad band absorption and atmospheric scattering (Platt and Stutz, 2008)" (or something like that)

③ page 2 line 59 "The named Multi-AXis-Differential Optical Absorption Spectroscopy (MAX-DOAS) instrument is designed …"-> "The Multi-AXis-Differential Optical Absorption Spectroscopy (MAX-DOAS) instrument is designed …"

④ page 3 line 69 "…trace gases concentrations…" -> "…trace gas concentrations…"

⑤ page 4 l98 "The cruise of ship-based observation" -> maybe just "The measurement cruise"

⑥ page 4 line99 "The ship-based measurements campaign" -> "The ship-based measurement campaign"

⑦ page 12 line254 "…impacted by the airflows transport…" -> "…impacted by airflow patterns…"

⑧ page 12 line258 "…showed the air masses were came from inland area…" -> "…showed the air masses coming from inland areas…"

⑨ line 272 "These two data sets time agreed well…" -> "These two data sets time agree well…"

R: We have followed these suggestions and corrected the mistakes accordingly.

**Reference:**

Fan, Q., Zhang, Y., Ma, W., Ma, H., Feng, J., Yu, Q., Yang, X., Ng, S.K.W., Fu, Q., Chen, L., 2016. Spatial and Seasonal Dynamics of Ship Emissions over the Yangtze River Delta and East China Sea and Their Potential Environmental Influence. Environmental Science & Technology 50, 1322-1329.

Fu, M., Liu, H., Jin, X., He, K., 2017. National- to port-level inventories of shipping emissions in China. Environmental Research Letters 12, 114024.

Hong, Q., Liu, C., Chan, K.L., Hu, Q., Xie, Z., Liu, H., Si, F., Liu, J., 2018. Ship-based MAX-DOAS measurements of tropospheric $NO_2$, $SO_2$, and HCHO distribution along the Yangtze River. Atmospheric Chemistry and Physics 18, 5931-5951.

Johansson, M., Galle, B., Yu, T., Tang, L., Chen, D., Li, H., Li, J.X., Zhang, Y., 2008. Quantification of total emission of air pollutants from Beijing using mobile mini-DOAS. Atmospheric Environment 42, 6926-6933.

Liu, H., Cheng, L., Xie, Z., Ying, L., Xin, H., Wang, S., Jin, X., Xie, P., 2016. A paradox for air pollution controlling in China revealed by "APEC Blue" and "Parade Blue". Scientific Reports 6, 34408.

Rodgers, C.D., 2000. Inverse methods for atmospheric sounding: theory and practice. World scientific.

Roscoe, H.K., Van Roozendael, M., Fayt, C., du Piesanie, A., Abuhassan, N., Adams, C., Akrami, M., Cede, A., Chong, J., Clémer, K., Friess, U., Gil Ojeda, M., Goutail, F., Graves, R., Griesfeller, A., Grossmann, K., Hemerijckx, G., Hendrick, F., Herman, J., Hermans, C., Irie, H., Johnston, P.V., Kanaya, Y., Kreher, K., Leigh, R., Merlaud, A., Mount, G.H., Navarro, M., Oetjen, H., Pazmino, A., Perez-Camacho, M., Peters, E., Pinardi, G., Puentedura, O., Richter, A., Schönhardt, A., Shaiganfar, R., Spinei, E., Strong, K., Takashima, H., Vlemmix, T., Vrekoussis, M., Wagner, T., Wittrock, F., Yela, M., Yilmaz, S., Boersma, F., Hains, J., Kroon, M., Piters, A., Kim, Y.J., 2010. Intercomparison of slant column measurements of $NO_2$ and $O_4$ by MAX-DOAS and zenith-sky UV and visible spectrometers. Atmospheric Measurement Techniques 3, 1629-1646.

Schreier, S.F., Peters, E., Richter, A., Lampel, J., Wittrock, F., Burrows, J.P., 2015. Ship-based MAX-DOAS measurements of tropospheric $NO_2$ and $SO_2$ in the South

China and Sulu Sea. Atmospheric Environment 102, 331-343.

Shaiganfar, R., Beirle, S., Denier van der Gon, H., Jonkers, S., Kuenen, J., Petetin, H., Zhang, Q., Beekmann, M., Wagner, T., 2017. Estimation of the Paris $NO_x$ emissions from mobile MAX-DOAS observations and CHIMERE model simulations during the MEGAPOLI campaign using the closed integral method. Atmospheric Chemistry and Physics 17, 7853-7890.

Shaiganfar, R., Beirle, S., Sharma, M., Chauhan, A., Singh, R.P., Wagner, T., 2011. Estimation of $NO_x$ emissions from Delhi using Car MAX-DOAS observations and comparison with OMI satellite data. Atmospheric Chemistry and Physics 11, 10871-10887.

Wang, S., Zhou, B., Wang, Z., Yang, S., Hao, N., Valks, P., Trautmann, T., Chen, L., 2012. Remote sensing of $NO_2$ emission from the central urban area of Shanghai (China) using the mobile DOAS technique. Journal of Geophysical Research 117 (D13305).

Wu, F., Xie, P., Li, A., Mou, F., Chen, H., Zhu, Y., Zhu, T., Liu, J., Liu, W., 2018. Investigations of temporal and spatial distribution of precursors $SO_2$ and $NO_2$ vertical columns in the North China Plain using mobile DOAS. Atmospheric Chemistry and Physics 18, 1535-1554.

Wu, F.C., Xie, P.H., Li, A., Chan, K.L., Hartl, A., Wang, Y., Si, F.Q., Zeng, Y., Qin, M., Xu, J., Liu, J.G., Liu, W.Q., Wenig, M., 2013. Observations of $SO_2$ and $NO_2$ by mobile DOAS in the Guangzhou eastern area during the Asian Games 2010. Atmospheric Measurement Techniques 6, 2277-2292.

Xing, C., Liu, C., Wang, S., Chan, K.L., Gao, Y., Huang, X., Su, W., Zhang, C., Dong, Y., Fan, G., Zhang, T., Chen, Z., Hu, Q., Su, H., Xie, Z., Liu, J., 2017. Observations of the vertical distributions of summertime atmospheric pollutants and the corresponding ozone production in Shanghai, China. Atmospheric Chemistry and Physics 17, 14275-14289.

---

## Author Comment (AC2) · 29 Sep 2018

We really appreciate the reviewers for the valuable and constructive comments, which are very useful for the improvement of the manuscript. We have replied the reviewers' comments point-to-point in below. The reviewers' comments are cited in black, while the responses are in blue. All the line number are referred to the revised manuscript.

(1) What's the purpose of comparison between ground measurements and satellite retrieval hereïij§In line 396 of section 4, the authors said "In order to validate the ship-based MAX-DOAS measurements…", however, satellite retrieval of trace gases have considerable uncertainties.

R: The unsuitable expression may lead to the confusion. The comparison between ship-based measurements and satellite retrieval can validate with each other. The ground-based data were commonly compared with satellite products in previous studies, however, the satellite products over marine areas were rarely validated by ground-based methods. In this study, the comparison between ship-based measurements and satellite data are aiming to provide validation of spaced observation over marine areas. We have re-phrased the sentence. Please refer to Line 418 to 420.

(2) Line 267, 10 km radius of location of ship-based measurements is selected to match with satellite pixel, but the satellite pixel especially the OMPS is much larger than this scope, how realize it?

R: The size of OMPS pixel is about $50 \times 50$ km$^2$, which is indeed too large for the comparison. In this study, the OMPS satellite products were firstly gridded in a high spatial resolution of 0.05°×0.05°, as described in Line 299 of the manuscript. For the typical cruising speed of 5 m/s, the travel distance is about 18 km within one hour (13:00-14:00 LT cover the satellites overpass time). To keep the consistency of temporal and spatial coverages, the area with the center of central longitude and latitude of ship-based measurements between 13:00 and 14:00 LT and radius of 10 km was chosen to average the satellite results.

(3) The current data analysis did not well support the authors' conclusion robustly.

Daily satellite observation can provide regional view of the distribution of gaseous pollutants, why the authors only show monthly data? How did the daily satellite data compare with of daily values of in the track of ship-based measurements? To reveal the transport and air pollution over sea, typical daily case is suggested.

R: Considering the ship-based measurements were carried out for almost a whole month from 2 to 29 June 2017 and satellite data are occasionally absent for some days, we presented the monthly averaged satellite results to reveal the general spatial distributions during the ship-based measurements. For the daily comparison, we presented the time series and correlation analysis of the satellite data and ship-based measurements in Figure 6 to 8 of the manuscript. High correlation coefficient R of 0.83, 0.76 and 0.69 were reported for $NO_2$, $SO_2$, and HCHO, respectively.

To follow the suggestion, some typical daily cases of $NO_2$ VCDs observed by these two instruments were shown in Figure R1. These two measurements and wind filed (black arrow) were overlapping plotted together, where the trajectories of ship-based measurements were indicated with the white lines and the central position of ship-based measurement during 13:00 to 14:00 LT were marked by black points.

It can be observed that these two data sets were highly consistent in the spatial distributions. Combining with the wind direction information, the air masses came from clean sea areas on 2 and 16 June, whereas originated from polluted inland areas on 7 and 27 June. Therefore, the observed $NO_2$ VCDs are substantially lower on 2 and 16 June compared to measurements on 7 and 27 June. Under the cleaner air masses from sea area, the hot spots of $NO_2$ pollution on 2 and 16 June are mainly located in the inland areas and some oceanic areas with strong ship emissions, which is blamed to local emissions. When the wind came from the polluted continental area, the $NO_2$ pollution spread from inland to the sea areas close to the coastal line on 7 and 27 June. It suggests that the air quality over sea areas were significantly influenced by pollutants transported from inland areas and even for the sea areas far from the coastal line.

The discussion about the daily case of air pollution transports was also added in the manuscript. Please refer to Line 312 to 330.

[Figure]

*Figure R1. Comparison of OMI and ship-based measured NO₂ VCDs on (a) 2, (b )7, (c) 16, and (d) 27 June. The ship-based measurements were plotted overlap in the base map of OMI products, and the wind field were indicated with black arrows.*

(4) It is important for the authors to clarify and emphasize what's new in their work and what's their new finding?

R: The novelty of this study were highlighted in the introduction and conclusion parts of the manuscript. It can be briefly summarized as: the ship-based MAX-DOAS measurements were first performed in the Eastern China Sea (ECS) area, and the typical trace gases spatial distribution were characterized. Meanwhile, the ship-based measurements are compared with satellite productions which is useful to validate the satellite retrieval in marine areas. The spatial distribution of these pollutant gaseous suggests that the air quality of the marine boundary layer in the ECS are mainly impacted by the air masses originated from the polluted inland areas and the local ship

emissions. We also reported the vertical structure of $NO_2$, $SO_2$, and HCHO in the ECS area. High concentration pollutants were identified in the sea areas of important ports and channels, which is related to the shipping emissions. Combining with the on board $O_3$ lidar instrument, we have discussed the $O_3$ formation process over marine areas. This study provided further understanding of the main air pollutants in the marine boundary layer of the ECS area.